# Evaluation of Compressive and Permeability Behaviors of Trabecular-Like Porous Structure with Mixed Porosity Based on Mechanical Topology

**DOI:** 10.3390/jfb14010028

**Published:** 2023-01-03

**Authors:** Long Chao, Yangdong He, Jiasen Gu, Deqiao Xie, Youwen Yang, Lida Shen, Guofeng Wu, Lin Wang, Zongjun Tian

**Affiliations:** 1College of Mechanical and Electrical Engineering, Nanjing University of Aeronautics and Astronautics, Nanjing 210016, China; 2College of Mechanical and Electrical Engineering, Jiangxi University of Science and Technology, Ganzhou 341000, China; 3Stomatological Digital Engineering Center, Nanjing Stomatological Hospital, Nanjing 210008, China; 4Nanjing Hangpu Machinery Technology Co., Ltd., Nanjing 211806, China

**Keywords:** trabecular-like porous structure, mixed porosity, mechanical topology, permeability, compressive strength, elastic modulus

## Abstract

The mechanical properties and permeability properties of artificial bone implants have high-level requirements. A method for the design of trabecular-like porous structure (TLPS) with mixed porosity is proposed based on the study of the mechanical and permeability characteristics of natural bone. With this technique, the morphology and density of internal porous structures can be adjusted, depending on the implantation requirements, to meet the mechanical and permeability requirements of natural bone. The design parameters mainly include the seed points, topology optimization coefficient, load value, irregularity, and scaling factor. Characteristic parameters primarily include porosity and pore size distribution. Statistical methods are used to analyze the relationship between design parameters and characteristic parameters for precise TLPS design and thereby provide a theoretical basis and guidance. TLPS scaffolds were prepared by selective laser melting technology. First, TLPS under different design parameters were analyzed using the finite element method and permeability simulation. The results were then verified by quasistatic compression and cell experiments. The scaling factor and topology optimization coefficient were found to largely affect the mechanical and permeability properties of the TLPS. The corresponding compressive strength reached 270–580 MPa; the elastic modulus ranged between 6.43 and 9.716 GPa, and permeability was 0.6 × 10^−9^–21 × 10^−9^; these results were better than the mechanical properties and permeability of natural bone. Thus, TLPS can effectively improve the success rate of bone implantation, which provides an effective theory and application basis for bone implantation.

## 1. Introduction

With the acceleration of population aging and the frequency of traffic accidents, bone defects have become a subject of serious concern in bone tissue engineering [1]. Suitable artificial implants have become the first choice of treatment, given the considerably long self-repair cycle of human bones and the limited capacity for reconstruction. The most common structure used in artificial implants is a porous one [2,3,4]. The introduction of porous features can reduce the apparent elastic modulus of artificial implants to a level close to that of the human bone (10 GPa), substantially reducing the occurrence of “stress shielding [5,6]. In addition, porous structures as artificial implants present several advantages, including a large specific surface area, good penetration performance, and high strength. Compared with other types of artificial implants, a reasonable porous structure design can promote the adhesion and proliferation of bone tissue cells on the scaffold, as well as realize the matching of penetration between the host bone and artificial implants. With the development of additive manufacturing technology (AM), the direct manufacturing of ultrahigh precision porous scaffolds has rather matured [7,8,9,10]. In order to achieve the unification of mechanical properties and permeability properties of implants, TI-6AL-4V porous scaffolds prepared by SLM technology are among the common artificial prostheses in bone tissue engineering [11,12].

In orthopedics, porous scaffolds can be divided into regular and irregular porous structures, depending on their basic structural units. Traditional research on artificial implants mainly focuses on regular porous structures, which are easy to optimize with respect to size and process. Regular internal filling structures, such as repeated unit cells, can be used to obtain lightweight structures. Common regular porous structures include diamond structures, regular hexahedron structures, and honeycomb structures [1,13,14,15,16,17]; however, regular porous structures are prone to stress shielding and exhibit poor permeability. An increasing number of studies have been conducted on irregular porous structures simulating the complex shape of real trabecular-like bone, and irregular porous scaffolds represented by Voronoi tessellation have drawn research interest [18,19,20]. Wang et al. based a parametric design of porous bone implants on the Voronoi subdivision [21,22,23,24,25]. The mathematical expressions between the modeling parameters of porous implants and the actual structural parameters of porous scaffolds were established using mathematical methods. Zhu et al. evaluated the effect of heat treatment on Voronoi porous titanium scaffolds and analyzed the fatigue properties of porous titanium scaffolds. Wang, Yiqiang et al. reconstructed irregular porous scaffolds of bone-like trabecular-like via computed tomography and studied the permeability behavior of porous titanium [18,26]. Liang, Huixin et al. conducted an in-depth study on the compression properties and the in vitro osteoblastogenesis of Voronoi porous scaffolds prepared by SLM (selective laser melting). Voronoi tessellation endows porous structures with a random combination of large and small holes and variations in pore shapes, as well as provides diversified and positive induction for the proliferation and differentiation of osteoblasts [25,27,28,29].

The compressive yield strength and elastic modulus of human bone tissue are close to 190 MPa and 10–30 GPa [1], respectively. Those with general porous titanium alloy parts are quite different from those of human bone tissue, inevitably producing stress shielding between porous implants and bone tissue. Due to the traditional porous structure having a relatively fixed pore size, it cannot control the pore size according to the requirements of the implant, and it is easy to fracture, and the bone penetration depth is low, so it is necessary to study a new porous structure design method [13,30,31,32]. In the current study, finite element analysis and the topology optimization of the designed model are first conducted. The pores of the designed trabecular-like porous structure (TLPS) are controlled based on the topology optimization results to achieve consistency between the mixed pores and the mechanical topology optimization results. In addition to the pore diameter, the pore shape can be controlled as well. This approach is biomimetic, controllable, and random; in addition, it is topological, which is a crucial problem affecting performance. The design method involves numerous characteristic parameters. The main design parameters, which include topology optimization, random points, scaling factor, load value, and irregularity, are combined with the application background of medical implants. Two basic characteristic parameters (porosity and pore size distribution), which are closely related to the physical and permeability properties, were selected for this study. On the basis of the representation of design parameters, this study discusses the relationship between design parameters and characteristic parameters by using statistical methods. Finally, the mechanical and permeability characteristics of TLPS validation are examined using compression and cell experiments, which provide an effective theoretical and applied research basis for the biomimetic design of implanted scaffolds.

## 2. Materials and Methods

### 2.1. TLPS Design

In the mechanical topology-based design process of trabecular-like porous structure with mixed porosity in Grasshopper, using the relative density mapping method and weighted random sampling method allows the control of the pore diameter and pore distribution of TLPS. In addition, the position and the density distribution of pores are optimized by finite element analysis and topology optimization. The principal stress line in the topology optimization area is extracted as a relative density mapping basis, and the pores are controlled using the relative density mapping principle. Therefore, the pores of the TLPS possess mixed controllable properties. Meanwhile, the shape and performance control problems under the synergistic constraints of mechanics, geometry, and manufacturing processes are comprehensively considered. The specific design methods in Grasshopper are as follows: (1) First, finite element and topology optimization analyses of the designed model are conducted, and the principal stress lines and topology optimization regions are obtained. (2) The principal stress lines in the topology optimization area are extracted. (3) The probability matrix is arranged, and the probability matrix is used to generate the seed points; the distance from the seed points to the principal stress line is calculated; the weighted random sampling method is used to retain the seed points, and the pore density distribution is controlled based on the density distribution of the seed points. (4) The scaling coefficient is set; the distance between the seed point and the principal stress line is mapped into the scaling coefficient; the corresponding scaling coefficient of each point is determined to control the scaling coefficient of each pore, realize the mix change of each pore size according to the distance between the point and the principal stress line, and generate TLPS. The aforementioned process is illustrated in Figure 1.

### 2.2. Mechanical Finite Element Analysis and Compression Testing

The commercial finite element software ABAQUS 6.14 (https://www.3ds.com/zh-hans/search/?wockw=abaqus) is used to simulate the structure to analyze the influence of design parameters on the mechanical properties of the TLPS. First, the model built in the software Grasshopper 6.0 (https://www.rhino3d.com/) is exported in STL format, imported into the software 3-Matic (www.materialise.com) to generate a volume grid, and exported in INP format. Ultimately, the file is imported into ABAQUS. The upper part of the model is moved down at a speed of 1 mm/min, and the bottom part of the model is fully constrained. The simulation environment during simulation is kept static and general, and all models are set based on the properties of the TI-6AL-4V material. The material properties are as follows: elastic modulus, 110.6 GPa; density, 4.4 × 10^−9^; and Poisson’s ratio, 0.326.

The TI-6AL-4V material exhibits excellent mechanical properties, biocompatibility, and corrosion resistance; as such, it is widely used in the application of orthopedic implants. Before the compression experiment, the designed TLPS are prepared using an SLM laser printer (Nanjing Chenglian, Nanjing, China). The printed material is commercial TI-6AL-4V powder, with power set to 160 W and a laser scanning speed set to 1250 mm/s supplied by EOS LTD (Shenzhen, China). Compression testing of the specimens was performed on a mechanical testing machine (CMT5105, MTS Systems, Shanghai, China). The crosshead displacement speed was fixed at 1 mm/min.

### 2.3. Permeability Simulation

The COMSOL 6.0 software (https://www.comsol.com/sla) is used to simulate it during the analysis of permeability characteristics. The watershed simulation model is first obtained by removing the porous structure via a Boolean operation and is then saved in STL format. After volume meshing with 3-MATIC software (www.materialise.com), the parts are imported into COMSOL, and the model is analyzed using the peristaltic flow model. For simplification of the simulation calculation and analysis, the deformation of the metal support is ignored as the fluid flows. The watershed material used is water, with the following properties: temperature, 37 °C (normal human body temperature); density, 1000 kg/m^3^; viscosity, 1.45 × 10^−9^ Mpa/s. The inlet velocity applied to the bracket is 1 mm/s, and the pressure at the outlet is zero. Under the assumption of a no-slip condition, Reynolds number in computational fluid dynamics is typically used to determine the fluid state. The object of the analysis is an incompressible fluid with constant density and thus is defined by the Navel–Stokes equation, as shown in Equation (2) [33,34,35].
(1)$$\rho \frac{\partial u}{\partial t}-\mu \nabla^{2}u+\rho \left(u\cdot \nabla \right)u+\nabla p=F$$
where *ρ* is the density of the fluid (kg/m^3^), *u* is the speed of the fluid (m/s), *t* is time (s), ∇ is the operator, *p* is the pressure (MPa), *μ* is the dynamic viscosity coefficient of the fluid, and *F* is the force (N).

The results report on the pressure drop, porosity, outlet flow rate, and permeability between the inlet and outlet surfaces of the watershed. The permeability was determined using Formula (2) of Darcy’s law, and the pressure gradient was measured using Formula (3) [32].
(2)$$K=v_{D}\cdot \mu_{d}\cdot \left(\frac{L}{\Delta P_{i-0}}\right)$$
(3)$$\Delta P=P_{Inlet}-P_{Outlet}$$
where *K* is the permeability, *v_D_* is the Darcy velocity, *u_d_* is the dynamic velocity, *L* is the model length, and *P* is the pressure gradient of the fluid domain.

### 2.4. Cell Experiment

The prepared TI-6AL-4V scaffolds were washed and activated prior to cell culture. The TLPS scaffolds were first sonicated in 95% alcohol and distilled water for 30 min. The TLPS scaffolds were then submerged in a 5 M sodium hydroxide solution at 60 °C for 24 h. They were further sonicated in distilled water for 10 min and then dried for 24 h to stabilize the oxide layer. The TLPS scaffolds were subjected to high temperature and autoclave sterilization.

Mouse osteoblasts (Mc3t3-e1 Cell Bank, Chinese Academy of Sciences, Beijing, China) were used to evaluate the cell permeability of the TLPS scaffolds. The cell culture temperature was set to 37 °C, and the environment contained 5% CO_2_. In addition, the medium used for cell culture was α-MEM (containing 10% fetal bovine serum and 3% penicillin–streptomycin, Gibco, Gaithersburg, MD, USA). Cell viability during culture was ensured by placing sterilized TLPS scaffolds into 24-well plates and adding the cell suspension (cell concentration 104 cells/mL) to each well of the plate, with the medium changed every 2 d. During the experiment, fluorescence live/dead staining was used to examine cell activity. The culture medium in the well plate was drawn out; the scaffold was washed with PBS 3 times, and the cells were fixed with 4% paraformaldehyde for 10 min. After the cells were washed with PBS, 0.1% Triton X-100 solution was added to them to increase cell permeability for subsequent staining. The cells were again washed twice with PBS and then stained for 15 min by adding a gophpene-cyclopeptide dye solution. The cells were stained for 3–5 min by adding a DAPI dye solution after absorption and then washed three times with PBS to wash off the excess dye solution. Finally, fluorescence detection was performed using a confocal microscope. In the experiment, SEM (scanning electron microscope) was used to observe the adhesion of cells on the surface of the scaffolds. First, the culture medium was removed after the cells were cultured for 4 d, and the scaffold was cleaned with PBS. Subsequently, 1 mL of 3% glutaraldehyde solution was added to each well and stored in a refrigerator at 4 °C overnight. The TLPS scaffolds were then immersed in gradient ethanol concentrations of 30%, 50%, 70%, 85%, 90%, 100%, and 100% for 15 min each, and the cells were dehydrated. The TLPS scaffolds were placed in a refrigerator at −80 °C overnight, and the cells were placed in a vacuum freeze dryer for 6–8 h to dry completely. The adhesion of cells on the TLPS scaffold was ultimately observed using a scanning electron microscope.

## 3. Results and Discussion

### 3.1. Controllability Analysis of Porosity and Pore Size Distribution

Studies have shown that the basic requirement for a natural bone to ensure the smooth transport and exchange of nutrients and metabolites is a pore size larger than 200 µm. Various researchers hold different views on the optimal shaping aperture range [4,15]; however, scaffolds with diameters of 600–1300 µm can generally meet the needs of bone tissue material transport and bone growth. In accordance with the structural design method, the main factors affecting the distribution of porosity and pore size include the number of random points N, topology optimization coefficient T, load value F, irregularity R, and scaling factor C. In the following, the single-variable method was used to analyze the five design parameters, and the statistical method is used to identify the sensitive factors affecting the porosity and pore size. The specific expressions between the porosity and pore size and other design factors are fitted using the experimental data, providing a theoretical basis and guidance for accurate TLPS design.

#### 3.1.1. Relationship between Design Parameters and Pore Size Distribution and Porosity

To evaluate the effect of design parameters (scaling factor C, topology optimization coefficient T, load value F, number of random points N, and irregularity R) on the characteristic parameters of porosity P and pore size distribution D, 25 groups of TLPS with different design parameters were designed. The pore size chosen for natural bone was 200–1000 µm, and the porosity selected was 40–90%. Therefore, the scaling factors chosen were 0.4–0.5, 0.4–0.6, 0.4–0.7, 0.4–0.8, and 0.4–0.9; the topology optimization coefficients chosen were 30%, 40%, 50%, 60%, and 70%; the load values selected were 1500 N, 2000 N, 2500 N, 3000 N, and 3500 N. The number of random points was 500, 600, 700, 800, and 900, and the degrees of irregularity was set at 0.4, 0.5, 0.6, 0.7, and 0.8. The parameter design is listed in Table 1.

In accordance with the design method, the distribution of porosity and pore size can be controlled by adjusting the scaling factor C, topology optimization coefficient T, load value F, number of random points N, and irregularity R. In determining the design parameters that largely influence the pore size distribution and porosity, five sets of C were designed such that other design parameters remained unchanged (Figure 2B). In the figure, as C varies, the pore size distribution of the TLPS ranges from 100 μm to 1400 μm; when C is from 0.4 to 0.5, the pore size is concentrated within the 100–400 μm range (Stage I), indicating that the TLPS has a small pore size and exhibits poor permeability. When C is between 0.4 and 0.6, the pore size distribution is concentrated within the 100–600 μm range (Stage I). This pore size distribution is conducive to cell adhesion but only slightly affects the transport ability. When C ranges from 0.4 to 0.7 (Stage II), the pore size distribution varies from 100 μm to 800 μm, and the pore size is less than 1200 μm of cancellous bone, which does not facilitate nutrient transport. When C ranges from 0.4 to 0.8 (Stages II/III), the pore size distribution region is from 100 μm to 1200 μm, which is consistent with the pore size distribution range of natural bone. When the scaling gradient is 0.4–0.9 (Stages II–III), the pore size distribution range exceeds 1200 μm, which is not conducive to cell adhesion. Therefore, C is set to 0.4–0.8 when other design parameters are analyzed. When other design parameters remain constant, the irregularity coefficient R and the topology optimization coefficient T are not; its correspondence with the pore size distribution of the TLPS is relatively close (Figure 2A,C). Within Stage I, its pore size distribution ranges from 100 μm to 700 μm. The scope is advantageous to cell growth; within Stage II, the pore size distribution is in the 700 to 1000 μm range, which is the transmission range of nutrition. In region III, the pore size distribution is 1000–1500 μm, corresponding to good permeability. In Regions I–III, the pore size distribution range corresponding to different values of R and T is relatively consistent, which only slightly affects the pore size distribution of the porous structure. When the other design parameters are unchanged, the load value F and the number of random points N vary. The pore size distribution stages I–Ⅲ are also relatively consistent, mainly distributed between 100 and 800 μm. The distribution is less than 800–1000 μm, and very few pores are larger than 1000 μm. Although the disturbance range of the pore size frequency curve is different, the overall distribution trend remains the same, and disturbance mainly occurs within the 100 to 1000 μm range. This result is with the requirements for natural bone implants. The pore size distribution of the TLPS conforms to the pore size distribution rule of the natural bone. Meanwhile, the load value F only slightly influenced the pore size distribution. As shown in Figure 2A–E, the main design parameter affecting the pore size distribution is the scaling factor C.

In addition to the pore size distribution, porosity affected the mechanical and permeability characteristics of TLPS. The greater the porosity and permeability of TLPS, the better, but the mechanical strength is lower. Thus, in TLPS design, the porosity control must fall within the appropriate range, which is consistent with the 50–90% porosity of natural bone. The effect of design parameters on the porosity of the TLPS was evaluated. The change process of porosity corresponding to five groups of design parameters is illustrated in Figure 3A–E. When other design parameters remained unchanged and the irregularity increased, the corresponding porosity first increased and then decreased; when the irregularity was 0.6, the maximum value of porosity was 65.6%; when the irregularity was 0.8, the minimum value of porosity was 64.9%. The porosity decreased by 0.5%, indicating that the irregularity only slightly affected the porosity. The main reason was that, with an increase in irregularity, the distribution area of random points was enlarged, and the corresponding pore edge tilt angle increased; however, the tilt angle exerted no effect on the pore size, hence the mild effect of the irregularity on the porosity. When other design parameters were unchanged and the number of random points increased, the porosity of TLPS initially decreased and then increased with an increase in the number of random points. When the number of random points reached 700, the porosity reached the minimum value of 65.6%; when the number of random points was 800, the porosity reached the maximum value of 66.7%, reflecting an increase of 1.1%. These results showed that the random points exerted a slight effect on the porosity mainly because, with an increase in the number of random points, the corresponding pore density increased, and with the scaling coefficient as a constant, the pore density increased, leading to a smaller pore edge diameter and a correspondingly larger pore size. The pore size was smaller compared with the seed points but greater than the porosity. When other design parameters remained unchanged and the load F increased, the corresponding porosity first decreased and then increased. The minimum value reached 65.1% when the load was 3000, and the maximum value reached 66.5% when the load was 3500, reflecting a small increase of 0.2%, indicating that the load only slightly influenced the porosity. The reason is that with an increase in load, the number of principal stress lines in the area of topology optimization increased. Moreover, the number of principal stress lines was reduced in the topology optimization area outside, leading to an increase in the seed point density in the topology optimization area. The corresponding pore size decreased, and the seed point density decreased in the topological optimization area outside; moreover, the corresponding pore size increased, retaining the average porosity and leading to a smaller change in overall porosity. When other design parameters remained constant, porosity increased with the topology optimization coefficient T. When the topology optimization coefficient was 30%, the porosity reached 64%; when the topology optimization coefficient was 60%, the maximum porosity was 67.3% and increased by 4.9%. This result showed that the topology optimization coefficient of the influence of the porosity increased. The main reason is that with an increase in the topology optimization area, the density of principal stress lines in topological regions increased, the number of pores increased, and the corresponding porosity increased. When other design parameters remained constant, the scaling factor increased, and the corresponding porosity increased with an increase in the scaling factor.

When the scaling factor was 0.4–0.9, the maximum porosity was 76.1%, increased by 34.9%, and showed that the influence of the scaling factor on the porosity was larger. The main reason is that, when other design parameters were the same, the scaling factor increased, each pore edge diameter decreased, and the pore size increased. Thus, the corresponding porosity of each pore increased, leading to an increase in the total porosity. The scaling factor greatly affected porosity, as determined from Figure 3A–E. Therefore, in TLPS design, the scaling factor has to be reasonably addressed for the porosity to reach the optimal value that can meet the mechanical and permeability characteristics of natural bone.

#### 3.1.2. Relationship between Pore Size Distribution and Porosity

Porosity is the most important factor affecting the mechanical and permeability properties of porous structures, and pore size is the most direct and important functional parameter of porous structures as medical implants [21]. If the pore size is too large, osteocytes cannot adhere, the bone tissue is loose, and the mechanical strength is poor; if the pore size is too small, the osteocytes cannot grow, and the cell tissue fluid cannot be transported effectively. Only when the pore size is within a reasonable distribution range can it be beneficial to osseointegration and bone ingrowth, and the advantages of the porous structure can be fully exploited. Therefore, the controllability of porosity and pore size distribution directly determines the stability of mechanical properties and permeability properties of TLPS. That is, under the premise of controllable design parameters, such as irregularity, topology optimization coefficient, load value, number of random points, and scaling factor, the mechanical and permeability properties of the porous structure are also controlled, provided that the porosity and pore size distribution are controlled.

The pore size distribution falls within a certain range, impeding data analysis. Thus, the pore size distribution is characterized by the average aperture. The larger the average aperture, the wider the corresponding pore size distribution. It can be characterized by the relationship between the average pore size and porosity. The design method indicates that the pore size is mainly affected by the distance between the random points. Thus, the relationship of point spacing with porosity and average aperture needs to be established. As shown in Figure 4, both the average aperture and porosity increase with an increase in point spacing. In the first stage, the average aperture and porosity increase faster with an increase in point spacing, and the average aperture ranges from 450 μm to 600 μm, which is conducive to cell attachment. In the second stage, the average aperture and porosity increase steadily, and the average aperture ranges from 550 μm to 680 μm, which is conducive to cell attachment and nutrient transport. In the third stage, the average aperture and porosity increase faster, and the average aperture ranges from 600 μm to 900 μm, which is conducive to nutrient transport but affects cell adsorption. On the basis of the aforementioned analysis, point spacing not only greatly affects the pore size but also the porosity. Therefore, the porosity increases with an increase in the average aperture, which is almost consistent with the change trend of the average aperture. Accordingly, in the design of the structure, the porosity of the TLPS can be controlled by controlling the pore size distribution to endow the TLPS with excellent mechanical and permeability properties.

### 3.2. Mechanical Finite Element Result

In selecting a more suitable TLPS scaffold for implantation experiments, the mechanical properties of scaffolds with different structural forms need to be analyzed. This section demonstrates the use of the simulation software ABAQUS 6.14 (https://www.3ds.com/zh-hans/search/?wockw=abaqus) to simulate the mechanical properties of 25 groups of TLPS. The simulation test results are shown in Figure 5. Under similar physical conditions such as loads and constraints, the TLPS corresponding to different design parameters exhibit different conducted stresses. However, as the pore changes, the stress conditions also change. As shown in Figure 5 (1–5), with an increase in C, the corresponding stress gradually rises. When C is 0.4–0.9, the largest stress distribution is reached, and the compressive strength is low, rendering the TLPS prone to fracture. Figure 5 (6–25) shows that the stress distribution is relatively close, and the stress is concentrated in the node part. Thus, when a fracture occurs, it is observed at the node where the TLPS connecting rods are connected, and the randomness of the porous structure renders the TLPS fragile and easily broken when the pore and edge are under load. The stress is often more concentrated on the fragile and easily broken hole and edge via the transmission of the connecting rod. The force can be gradually transferred from the loading area to the layered slices at different levels. Comprehensive analysis indicates that the topology optimization coefficient, load value, number of random points, and irregularity only lightly affect stress distribution, whereas the scaling factor largely influences stress distribution. However, when the scaling factor is small, the compressive strength is large, and stress shielding is likely to occur. Therefore, the elastic modulus needs to be studied in a subsequent compression test.

For an in-depth analysis of the mechanical characteristics of the designed structure, the displacement variation trend (Figure 6) under similar conditions is investigated in this study. The displacement variation is relatively apparent in 1–5 groups, which also indicates that the scaling coefficient exerts the greatest mechanical influence on the TLPS. For the TLPS design, matching requirements of stress and modulus need to be satisfied, and rupture of the structure due to excessive stress after implantation should be avoided. Moreover, the metal debris that falls off exhibits certain toxicity, which is harmful to human health. For the design of lightweight porous structures, not only the rigidity and strength of the structure are required but also the stability, resistance, and energy absorption capacity of the overall structure. Therefore, this study focuses on the comparison of the elastoplastic stage result of the compression test for parts formed by SLM and the FEA model. This study also analyzes the mechanical curve after failure to explore its various properties in lightweight applications.

### 3.3. Compression Test Results

Mechanical properties are crucial in bone implant applications. When the TLPS exhibit mechanical properties within a certain range, the compressive strength of bone implants can be effectively improved, and stress shielding can be addressed to a certain extent [31,32,34,35]. Therefore, in TLPS design, the compressive strength and elastic modulus of the structure should be comprehensively considered.

In this study, the mechanical properties of bone implant TLPS are evaluated in two aspects: the apparent elastic modulus and the ultimate compressive strength. The apparent elastic modulus (E) is characterized by the quasi-elastic gradient (ISO13314:2011), and the compressive strength (S) is characterized by the ultimate compressive stress. The load–displacement curve obtained on the testing machine is transformed into an engineering stress–strain curve (Figure 7). As shown in the figure, nonlinearity is observed before the linear elastic phase of the curve. The reason is that the scaffold and indenter establish a state of complete contact during compression. The curves reveal that the TLPS scaffold shows no obvious yielding behavior. The compressive strength is characterized by the ultimate compressive stress, the elastic modulus of the scaffold is the slope of the elastic deformation stage of the stress–strain curve, and the compressive strength is the stress corresponding to the peak value of the stress–strain curve. Figure 7E–I lists the compression curves of the scaffold measured by uniaxial compression testing. Figure 7E,F show that as the scaling factor C increases, the stress value corresponding to the stress curve gradually decreases; in addition, as the topology optimization coefficient T increases, the corresponding stress increases, indicating that the TLPS are greatly affected by C and T. As shown in Figure 7G–I, the stress in the corresponding compression curve of the TLPS is affected by the load value F, the number of random points N, and the irregularity R. The effect is small, and its corresponding stress is almost unchanged.

Elastic modulus and compressive strength are two important parameters to characterize the mechanical properties of porous structures, and compressive strength determines the maximum load that the structure can bear. Figure 8 shows the variation trend of compressive strength corresponding to different design parameters. As shown in Figure 8A, when other design parameters remain unchanged, the compressive strength decreases with an increase in C. When C is 0.4–0.5, the compressive strength reaches the maximum value of 580 Mpa, while, with a C value of 0.4–0.9, the compressive strength decreases by 34.9%. The reason is that, with an increase in C, the corresponding pore size and edge diameter decrease. Under the condition that the number of pores remains unchanged and the compressive strength decreases, the compressive strength is consistent with the results of the finite element analysis. Thus, C significantly affects the compressive strength of the porous structure. When other design parameters remain unchanged and T increases gradually, the compressive strength increases with an increase in T. When T is 30%, the compressive strength reaches the minimum value of 270 MPa, while, with a T value of 70%, the compressive strength increases by 23%, mainly due to the increases in T and the topology area. The density of the principal stress line increases correspondingly, resulting in denser pores, thereby improving the compressive strength. However, compared with the effect of C on compressive strength, T is relatively small. As shown in Figure 8D, when other design parameters remain unchanged and N increases gradually, the corresponding compressive strength generally increases. When N is 500, the compressive strength reaches the minimum value of 320 MPa, while with, a N value of 800, the compressive strength increases by 15.7%; however, when N is 900, the corresponding compressive strength value decreases, mainly because, when N increases, the corresponding pores become denser, causing an increase in the compressive strength; however, when N reaches a certain value, the corresponding pores become too dense, resulting in smaller pore size and edge diameter, thereby decreasing the compressive strength. As shown in Figure 8C,E, when other conditions remain unchanged, the compressive strength is less affected by F and R, increasing by 2% and 3%, respectively. Thus, the effects of F and R on the mechanical properties are negligible.

The elastic modulus is an important parameter to characterize the mechanical properties of the TLPS, and its matching determines whether it can solve the stress shielding problem. Figure 9 shows the change trend of the elastic modulus corresponding to the TLPS under different design parameters, as shown in Figure 9A. When C increases, the elastic modulus gradually decreases. When C is 0.4–0.5, the elastic modulus reaches the maximum value of 9.716 GPa, while, with a C value of 0.4–0.9, the elastic modulus decreases by 25.1%. The main reason is that, under the action of the same force, the deformation of the TLPS gradually increases with an increase in C, which helps address stress shielding; however, a low elastic modulus can easily lead to TLPS fracture. As shown in Figure 9B–E, with other design parameters remaining constant, the elastic modulus of the TLPS initially increases and then decreases with increases in T, F, N, and R to 8.5%, 12.3%, 14.6%, and 26%, respectively. In addition, R largely affects the elastic modulus mainly because when R increases, the inclination angle of the pore edge gradually increases; in addition, when R is less than 0.6, the compressive strength corresponding to the TLPS gradually increases, indicating that the pore–rib inclination angle is less than 45°, which is conducive to improving the mechanical strength. Consequently, the elastic modulus gradually increases. When the irregularity exceeds 0.6, the corresponding compressive strength and elastic modulus decrease, indicating that the pore–rib inclination angle is greater than 45°. The TLPS begins to break, causing the elastic modulus to decrease. R should then be controlled within 0.6 in TLPS design. When N is greater than 800, the elastic modulus also begins to decrease, mainly because the number of pores gradually increases with an increase in N; when the number of pores reaches a certain level as C remains constant, the pore size and edge diameter become smaller, resulting in poor mechanical properties. The elastic modulus then decreases. When T is greater than 50%, the topology optimization area exceeds the non-topology optimization area. With N remaining unchanged, the pore size in the topology optimization area increases. Consequently, TLPS displacement increases, causing a decrease in the elastic modulus of the TLPS. When F is greater than 2500, with T remaining unchanged, the principal stress lines in the topological area become denser, the corresponding pore size becomes smaller, and the pore size in the non-topological optimization area becomes larger. As TLPS displacement increases, the elastic modulus decreases.

The aforementioned data suggest that the TI6AL4VTLPS scaffold meets the requirements of natural bone with respect to elastic modulus and has a compressive strength exceeding 190 MPa of natural bone, indicating that this TLPS design shows great potential in bone implantation applications. In TLPS design, the structure should have a good elastic modulus on the basis of improving the compressive strength to avoid stress shielding. The aforementioned analysis indicates that C largely affects the compressive strength and elastic modulus of the TLPSs. Therefore, in the permeability experiment design, the design parameters should be studied, together with the permeability characteristics, under different design parameters.

### 3.4. Permeability Analysis

Permeability represents the ability of fluids to flow through a porous structure and the ability to transport nutrients or metabolites in bone tissue. It is an important parameter for the in vivo performance of bone implants and is essential for cell adhesion and proliferation for vascularization. For porous scaffolds, proper permeability can promote tissue regeneration and improve the success of implantation. However, increasing permeability reduces the mechanical strength of TLPS scaffolds. Therefore, the rational selection of the internal structure of porous TI6AL4Vscaffolds is crucial for porous orthopedic implants and bone substitutes. However, design parameters such as C, T, F, N, and R determine the permeability. The effects of design parameters on the permeability of TLPS are assessed.

As shown in Figure 10, the fluid simulation analysis of 25 groups of TLPS is conducted. The pressure drop and permeability of the scaffolds need to be evaluated to evaluate the permeability properties of the scaffolds after implantation and thus analyze the transport properties of cells and nutrients between different pores. In Figure 10A,B (1–5), with an increase in the C value of the TLPS, the corresponding flow rate is larger, the pressure drop gradually decreases, and the permeability is improved. In Figure 10A,B (6–25), the fluid velocity distribution and pressure drop of the TLPS is relatively disordered, but the fluid velocity and motion trajectory are relatively consistent, it can also be seen from the Figure 10 that the pressure drop and velocity change law is opposite, which is consistent with Formula (2). The velocity increases initially and then decreases with an increase in T. Moreover, F N, R, and T affect the flow velocity of the TLPS; however, the effect is small compared with that of the scaling factor.

Under similar conditions, the pressure drop and permeability of the TLPS calculated using Formulas (2) and (3) are shown in Figure 11 and Figure 12. The permeability is almost consistent with the corresponding change law of the flow-rate cloud map. As shown in Figure 11A, the permeability increases with an increase in C. When C is 0.4–0.5, the permeability reaches the minimum value of 3.13 × 10^−9^, while, with a C value of 0.4–0.9, the permeability increases by 85%. The primary reason is that, when other design parameters remain unchanged, the pore size increases with an increase in C. The corresponding flow rate also increases, enhancing the permeability; however, although it facilitates nutrient transport, permeability, when it is too high, hampers cell attachment. As shown in Figure 11B, the permeability of the TLPS decreases with an increase in T by 52%, mainly because of the higher T of the TLPS when other design parameters remain unchanged. The larger the corresponding topological area, the denser the corresponding pores in this area, resulting in a lower flow rate and lower permeability. As shown in Figure 11C, with an increase in F, the permeability of the TLPS increases first and then decreases by 42%. The primary reason is that when other design parameters remain unchanged, the principal stress line density in the corresponding topology optimization area increases with an increase in F, thereby decreasing the pore size in this area. At this time, the pore size in the nontopological area increases, causing an increase in the flow rate, and the corresponding permeability increases; however, when F increases to a certain value, the pore density further increases, and the pore size further decreases. Consequently, the flow rate decreases, together with the permeability. As shown in Figure 11D, with an increase in N, the permeability of the TLPS decreases. In this instance, the decrease is 63%, which is mainly attributed to an increase in pore density as N increases, given that the other design parameters remain unchanged. Consequently, the pore size and flow rate decrease, reducing the permeability. As shown in Figure 11E, when other design parameters remain unchanged, the permeability is less affected by R, and the variation range is 21% because the irregularity mainly affects the pore shape and only slightly affects the pore size and pore density. Thus, the effect of R on permeability is smaller.

The effect on the fluid pressure drop of the TLPS under different design parameters is analyzed (Figure 12). The nutrient transport characteristics can be further analyzed by studying the change trend of the pressure drop. As shown in Figure 12, the change in the pressure drop of the fluid is inversely proportional to the permeability, which is consistent with Equation (3). As shown in Figure 12A–E, with an increase in C, the pressure drop decreases by 40%, given that the other design parameters remain unchanged. It decreases by 13% and 12% with changes in F and R, respectively, and decreases by 25% and 23% with changes in T and N, respectively. These results indicate that C greatly affects the pressure drop of the TLPS, whereas other design parameters only slightly influence the pressure drop of the TLPS, which is more consistent with the permeability analysis.

The permeability of human bones reported in the literature is 0.4–11.0 × 10^−9^. Meanwhile, the permeability coefficient of the TLPS is 0.6~2.1 × 10^−8^, as determined by COMSOL simulation calculation. As observed, the flow characteristics of the TLPS are consistent with those of cancellous bone, and the pore density distribution of the structure is uneven and controllable. These qualities facilitate the migration of cells and nutrient materials to the depth of the scaffold. The closer to the inner surface boundary, the lower the flow velocity. These conditions contribute to the adsorption of cells and nutrients on the inner surface of the scaffold. These substances are necessary for the growth of bone tissue and can promote the growth of subsequent bone tissue. Combining the effects of different design parameters on permeability and pressure drop, C exerts the most effect on the permeability and pressure drop. Therefore, in the analysis of permeability, the characteristics of permeability under different scaling factors need to be investigated.

### 3.5. Cell Testing

In order to analyze the effects of different design parameters on the permeability properties of TLPS scaffolds are analyzed. The aforementioned analysis of permeability and mechanical properties indicates that C exerts the most substantial effect on the mechanical and permeability properties. Therefore, five groups with different C values were selected for cell test analysis. Figure 13A–C respectively presents the SEM morphology, fluorescent staining results, and cell proliferation results of cells on the surface of the TLPS scaffolds.

As shown in Figure 13A, when C is 0.4–0.5, the number of cells attached to the scaffold surface is small. As shown in the SEM images, the cell spreading area on the scaffold surface is relatively small, the pseudopodia stretch less, and the overall shape is polygonal. No association is established between them, indicating that its permeability is weak, which is not conducive to the adhesion and spread of MC3T3-E1. As C increases, the number of cell pseudopodia protruding increases. They adhere to the edge of the groove to significantly increase the spreading area of the cells. The staining results showed that the cell arrangement and growth surface showed strong directionality, roughly in the direction of the pores. Different cells also aggregated to form cell groups. The aforementioned results show that, with an increase in C, the corresponding porosity increases, which markedly promotes the adhesion and growth of cells on the surface. However, when C is 0.4–0.9, the extension of the pseudopodia of the cells is reduced, the fluorescence staining results show that the number of cells is significantly reduced, and the cell activity is significantly reduced. These results show that with an increase in C, the porosity increases, and the corresponding permeability is considerably high, which is not conducive to cell adhesion and spreading. Compared with the other three groups, MC3T3-E1 shows a significantly larger spread on the surface area, and the marginal pseudopodia are more abundant. In addition, the cells are polygonal. The aforementioned results indicate that MC3T3-E1 is highly sensitive to the TLPS, and the pseudopodia protruding from cells can well adhere to the surface of the structure, thereby increasing the spreading area of the cells. These pseudopodia also help achieve cell migration, aggregation, and other functions. In the fluorescent staining results and cell proliferation results, the cells on the surface of C is 0.4–0.7 and 0.4–0.8 exhibit good agglomeration, and the cells are connected by filopodia. The cells continue to grow into a cell cluster, which is crucial for information exchange and gene expression between cells in later stages. The differences in the cultured cell densities of the five groups of scaffolds are mainly attributed to the differences in the local permeability of the TLPS, which can directly affect the seeding efficiency of cells on the scaffold. Higher permeability indicates less resistance for cell suspension to permeate the scaffold. It can better deliver nutrients and oxygen to the cells and avoid blockage of voids, which is favorable for the maintenance of cell viability and proliferation. However, when the permeability reaches a certain level, the time for cells to attach to the surface of the scaffold is short, which is not conducive to cell adhesion and spreading. With an increase in permeability within a certain range, the adhesion and spreading of MC3T3-E1 are significantly promoted, relative to those of the less permeable structure surface. This occurrence markedly reduces the risk of implants in service in the human body, which improves the success rate of the implant operation.

In order to further analyze the cell proliferation in each group of scaffolds with different permeability, OD value testing of five groups of TLPS scaffolds with different C values was performed. Figure 14 shows the OD values, the null group described the control group without any scaffold. On Day 1 of culture, the numbers of cells on the surface of the TLPS scaffold in each group hardly vary, and no significant difference exists. On Day 4, the proliferation level of cells in all groups increases, the number of cells in Groups 1, 2, and 5 are almost similar, and the cell proliferation levels in Groups 3 and 4 are significantly enhanced. These observations significantly vary from those in Groups 1, 2, and 5. In Group 3, the effect of the surface on cell proliferation is more apparent. On Day 7 of the culture, cell proliferation in each group reached improved levels. Among the groups, Group 3 and 4 shows a higher proliferation and a larger number of cells, confirming the positive effect of a higher permeability within a certain range on promoting cell proliferation; the number of cell proliferation in group 6 was the highest, indicating that the TI-6AL-4V scaffold had a certain degree of influence on cell proliferation.

The number of cells on Day 4 shows no significant increase relative to that on Day 1, with a corresponding increase of 17%. In addition, the number of cells on Day 7 shows a significant increase of 48%. This occurrence is mainly attributable to the classification of the cell cycle into four stages: proliferation, differentiation, mineralization, and apoptosis. With an extension of culture time, cells may enter the differentiation stage from the proliferation stage; at this time, the inhibition of cell proliferation occurs. Proper permeability is the primary prerequisite for ensuring bone ingrowth and cell proliferation and differentiation. It is also the most direct functional parameter of porous structures as medical implants. Moreover, the permeability of the scaffolds is affected by the porosity, and the porosity is mainly affected by C. Therefore, the larger the C value, the higher the permeability. When the permeability is too high, the bone cells cannot adhere. Moreover, the bone tissue is loose and the mechanical strength is poor; the smaller the C, the lower the permeability. Under these conditions, the osteocytes cannot grow, and the cell tissue fluid cannot be transported effectively. Appropriate permeability can significantly promote the proliferation of cells. In this experiment, changing the permeability of the scaffold also shows the potential to improve cell proliferation.

## 4. Conclusions

In this study, a novel method for the design of trabecular-like porous structure is proposed, and the mechanical strength and permeability of the structures with different design parameters are tested and analyzed. The permeability characteristics of the designed structure are verified by cell adhesion. The main conclusions are as follows:

(1) The proposed method for the design of trabecular-like porous structure is not only suitable for regular models but can also be used to design complex and irregular models, such as the acetabular cup, mandibular repair, femur, and so on, can be widely used in bone implants and bone repair. The relative density mapping and weighted random sampling strategy are used to adaptively control the pore distribution and pore size. Under the premise of establishing a parametric model, the porous structures can be adaptively generated by merely importing different model data and without any need for redundant iterations.

(2) The trabecula-like porous structures are mainly determined by design parameters, such as scaling factor, topology optimization factor, load value, number of random points, and irregularity, and so on. By adjusting these design parameters, the porosity and pore size distribution can be controlled. The scaling factor considerably influences the porosity, which is linear. Moreover, the porosity increases when the average pore size increases and is almost consistent with the change trend demonstrated by the average pore size.

(3) After mechanical finite element analysis and compression testing, the compressive strength and elastic modulus of trabecula-like porous structures are higher than those of natural bone, and the range can be controlled using the design parameters. The TLPS can effectively improve mechanical strength and help solve the stress shielding problem in bone implantation.

(4) Permeability simulation and cell experiments demonstrate that, when the scaling factor is 0.4–0.7 or 0.4–0.8, the penetration characteristics of trabecular-like porous structure are similar to those of natural bone; in addition, to a certain extent, the permeability of the TLPS is better than that of natural bone.

## Figures and Tables

**Figure 1 jfb-14-00028-f001:**
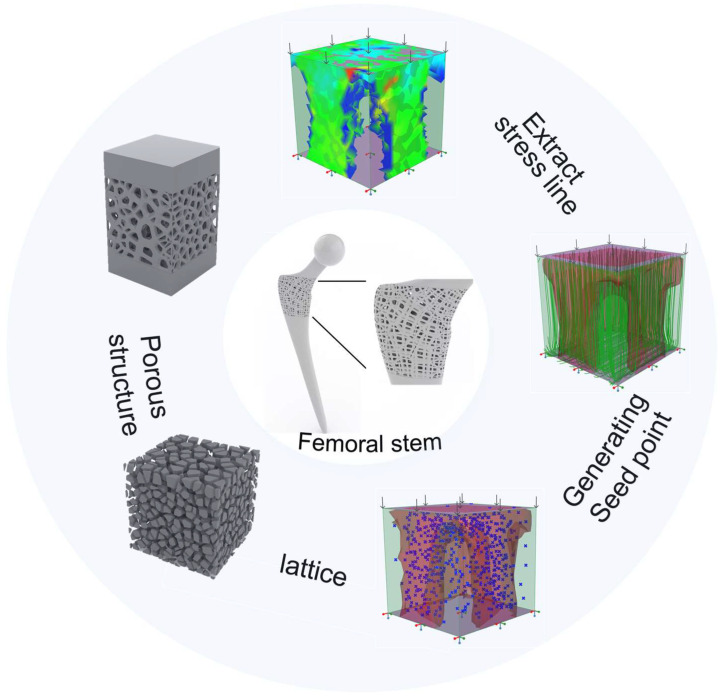
TLPS design in Grasshopper.

**Figure 2 jfb-14-00028-f002:**
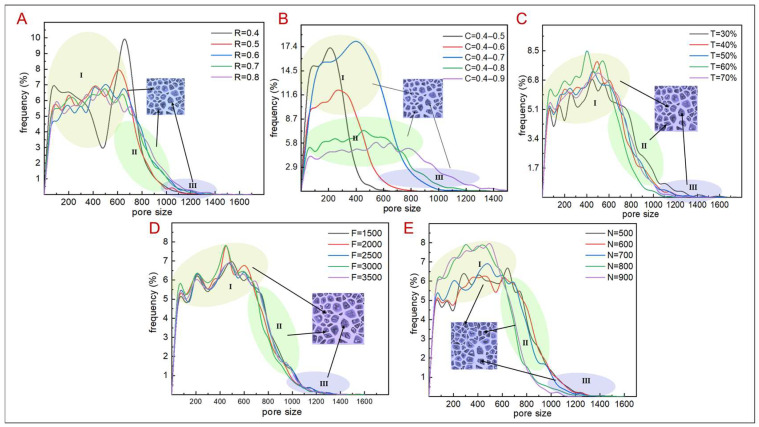
TLPS pore size distribution frequencies corresponding to different design parameters. (**A**) R corresponds to the pore size distribution. (**B**) C corresponds to the pore size distribution. (**C**) T corresponds to the pore size distribution. (**D**) F corresponds to the pore size distribution. (**E**) N corresponds to the pore size distribution. The “I” “II” and “III” respectively represent the pore size distribution range corresponding to different parameters.

**Figure 3 jfb-14-00028-f003:**
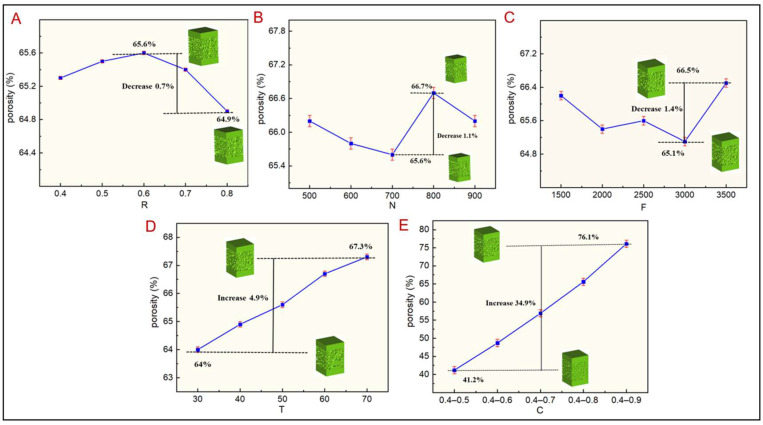
Variation trend of TLPS porosity corresponding to different design parameters. (**A**) R corresponds to the porosity. (**B**) N corresponds to the porosity. (**C**) F corresponds to the porosity. (**D**) T corresponds to the porosity. (**E**) C corresponds to the porosity. The green column represents the TLPS model under different parameters.

**Figure 4 jfb-14-00028-f004:**
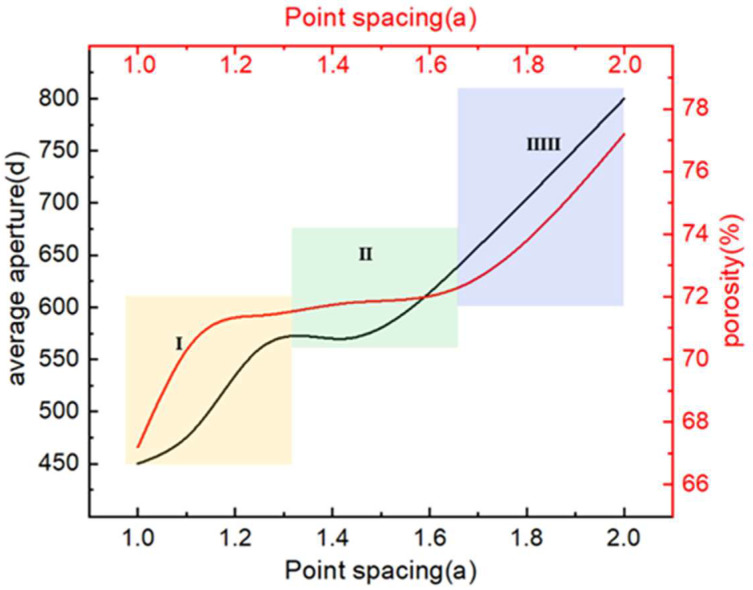
Relationship between porosity and pore size corresponding to TLPS. The “I” “II” and “III” respectively represent the variation trend of average aperture with different point spacing ranges.

**Figure 5 jfb-14-00028-f005:**
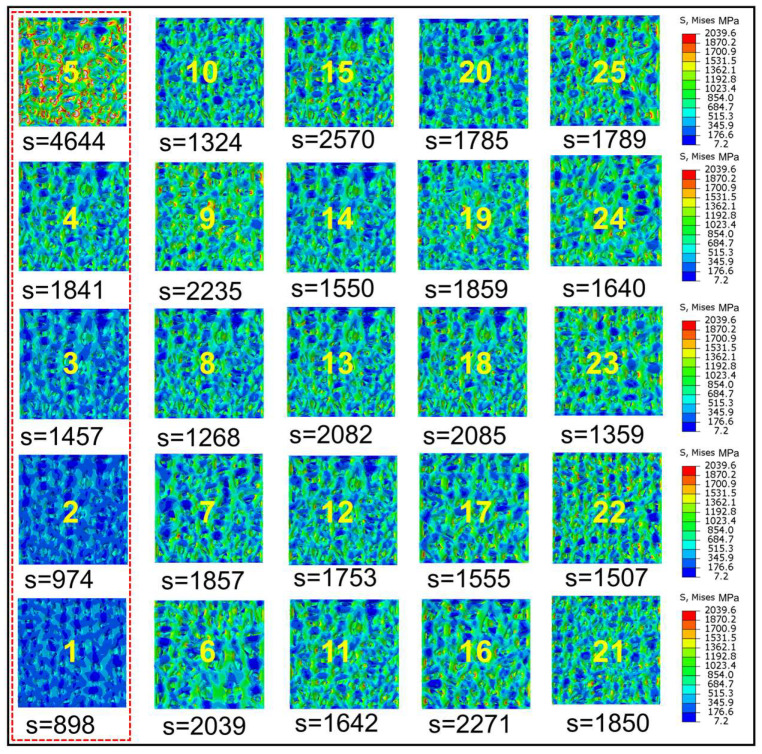
TLPS stress contour corresponding to different design parameters. The numbers 1–25 respectively represents the corresponding TLPS model number. S represents the maximum mises value.

**Figure 6 jfb-14-00028-f006:**
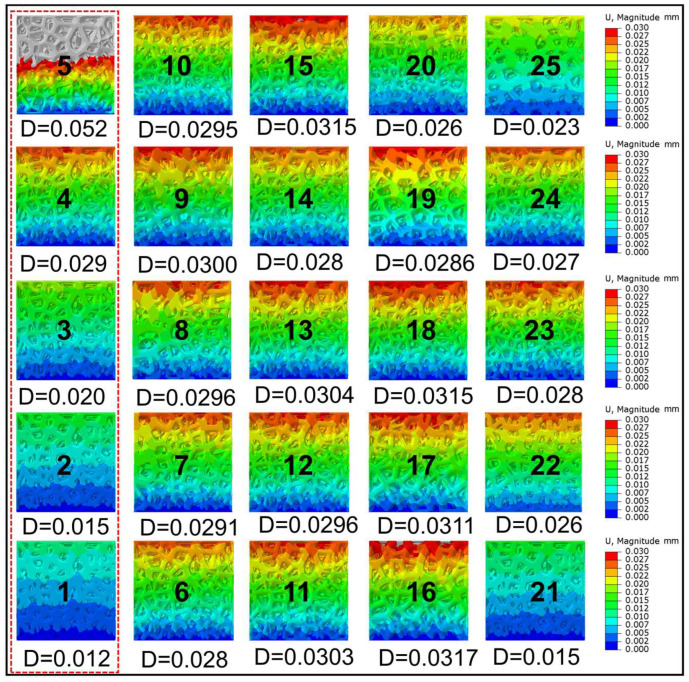
TLPS displacement contour corresponding to different design parameters. The numbers 1–25 respectively represents the corresponding TLPS model number. D represents the maximum displacement value.

**Figure 7 jfb-14-00028-f007:**
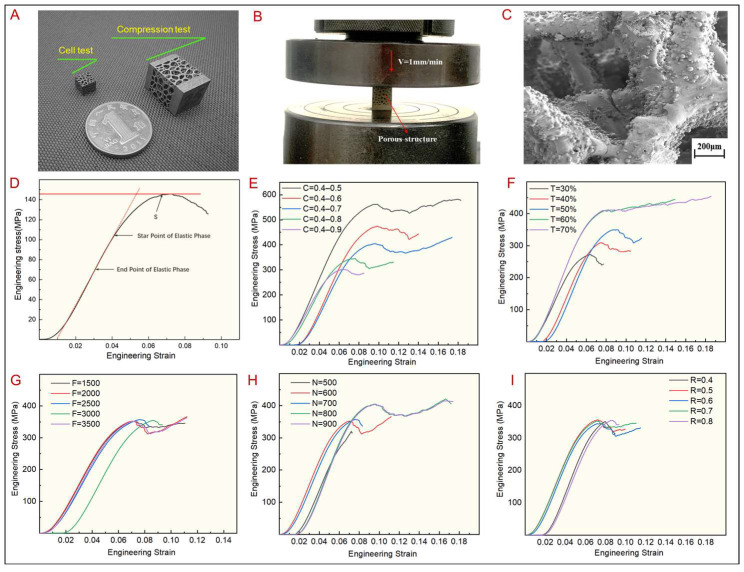
TLPS compression testing. (**A**) TLPS samples prepared by SLM. (**B**) TLPS compression process. (**C**) TLPS microstructur. (**D**–**I**) Stress-strain curves corresponding to different design parameters.

**Figure 8 jfb-14-00028-f008:**
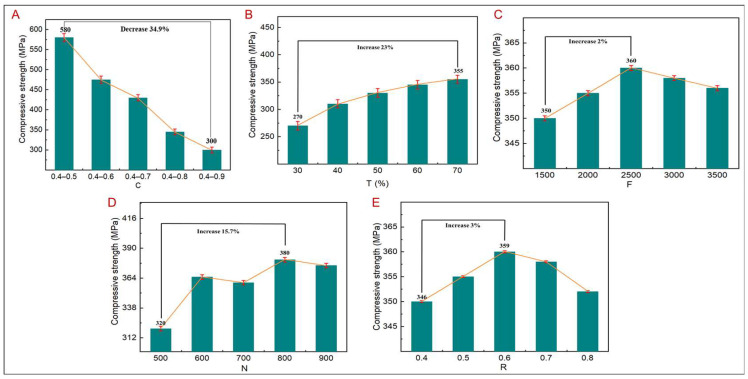
Compressive strength of TLPS corresponding to different design parameters. (**A**) C corresponds to compressive strength. (**B**) T corresponds to compressive strength. (**C**) F corresponds to compressive strength. (**D**) N corresponds to compressive strength. (**E**) R corresponds to compressive strength.

**Figure 9 jfb-14-00028-f009:**
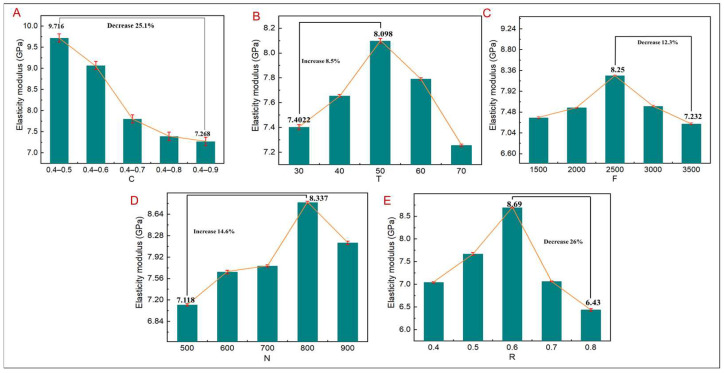
Elastic modulus of TLPS corresponding to different design parameters. (**A**) C corresponds to elasticity modulus. (**B**) T corresponds to elasticity modulus. (**C**) F corresponds to elasticity modulus. (**D**) N corresponds to elasticity modulus. (**E**) R corresponds to elasticity modulus.

**Figure 10 jfb-14-00028-f010:**
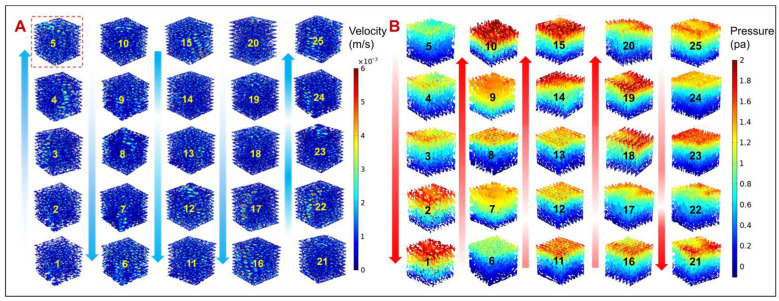
The fluid simulation analysis result of 25 groups of TLPS. (**A**) Velocity distribution of the TLPS flow field. Red dotted line box represents the model with the maximum flow velocity. (**B**) Pressure distribution of the TLPS flow field.

**Figure 11 jfb-14-00028-f011:**
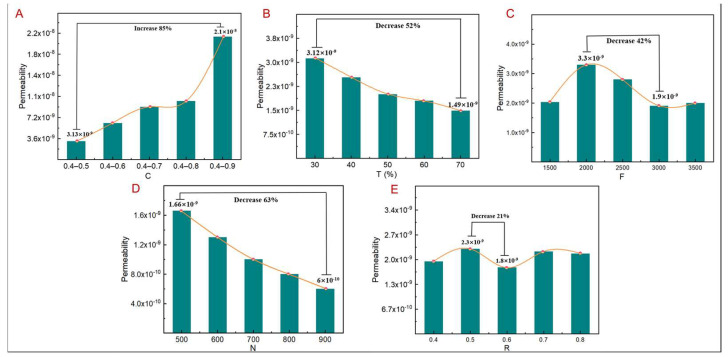
TLPS permeability corresponding to different design parameters. (**A**) C corresponds to permeability. (**B**) T corresponds to permeability. (**C**) F corresponds to permeability. (**D**) N corresponds to permeability. (**E**) R corresponds to permeability.

**Figure 12 jfb-14-00028-f012:**
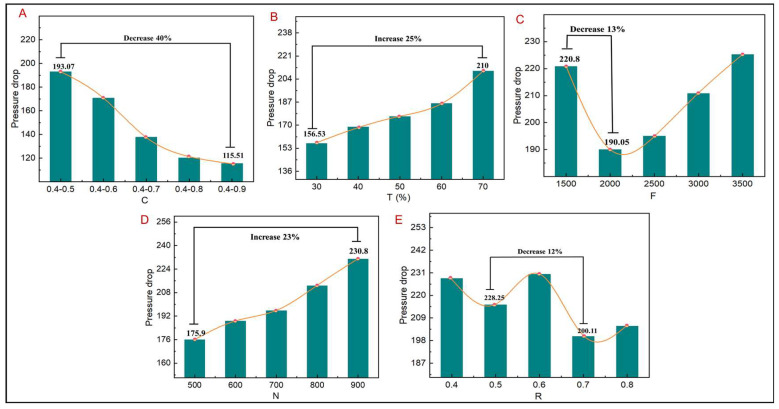
TLPS pressure drop corresponding to different design parameters. (**A**) C corresponds to pressure drop. (**B**) T corresponds to pressure drop. (**C**) F corresponds to pressure drop. (**D**) N corresponds to pressure drop. (**E**) R corresponds to pressure drop.

**Figure 13 jfb-14-00028-f013:**
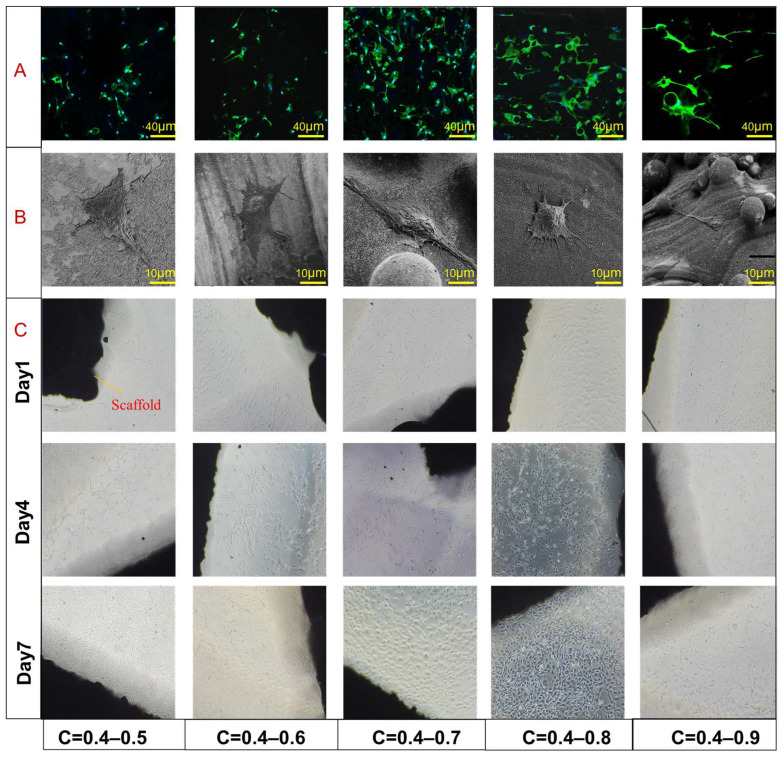
Cell test results. (**A**) TLPS fluorescence staining results. (**B**) TLPS SEM test result. (**C**) Cell proliferation results of cells on the surface of each group of TLPS scaffolds.

**Figure 14 jfb-14-00028-f014:**
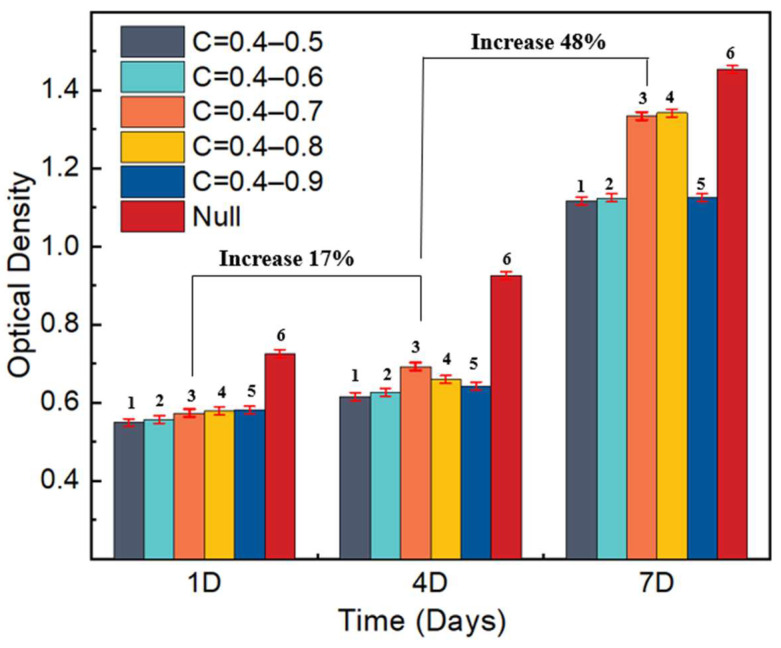
Proliferation conditions of MC3T3-E1 cells on scaffolds: variation of OD with C. (1 represents C = 0.4–0.5; 2 represents C = 0.4–0.6; 3 represents C = 0.4–0.7; 4 represents C = 0.4–0.8; 5 represents C = 0.4–0.9; 6 represents without samples).

**Table 1 jfb-14-00028-t001:** Design parameters.

Series	C	T	F	N	R
**1**	0.4–0.5	50%	2500	700	0.6
**2**	0.4–0.6	50%	2500	700	0.6
**3**	0.4–0.7	50%	2500	700	0.6
**4**	0.4–0.8	50%	2500	700	0.6
**5**	0.4–0.9	50%	2500	700	0.6
**6**	0.4–0.8	30%	2500	700	0.6
**7**	0.4–0.8	40%	2500	700	0.6
**8**	0.4–0.8	50%	2500	700	0.6
**9**	0.4–0.8	60%	2500	700	0.6
**10**	0.4–0.8	70%	2500	700	0.6
**11**	0.4–0.8	50%	1500	700	0.6
**12**	0.4–0.8	50%	2000	700	0.6
**13**	0.4–0.8	50%	2500	700	0.6
**14**	0.4–0.8	50%	3000	700	0.6
**15**	0.4–0.8	50%	3500	700	0.6
**16**	0.4–0.8	50%	2500	500	0.6
**17**	0.4–0.8	50%	2500	600	0.6
**18**	0.4–0.8	50%	2500	700	0.6
**19**	0.4–0.8	50%	2500	800	0.6
**20**	0.4–0.8	50%	2500	900	0.6
**21**	0.4–0.8	50%	2500	700	0.4
**22**	0.4–0.8	50%	2500	700	0.5
**23**	0.4–0.8	50%	2500	700	0.6
**24**	0.4–0.8	50%	2500	700	0.7
**25**	0.4–0.8	50%	2500	700	0.8

C stands for scaling factor. T stands for topology optimization coefficient. F stands for load value. N stands for number of random points. R stands for irregularity.

## Data Availability

Not applicable.

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
