# Peer review of "Evaluation of Compressive and Permeability Behaviors of Trabecular-Like Porous Structure with Mixed Porosity Based on Mechanical Topology"

_jfb, 2023, doi:10.3390/jfb14010028_

Round 1
Reviewer 1 Report
1. Add details about the matematical model for geometry generation from Grasshopper.
2. Mention the initial dimesions pore sizes.

Reviewer 2 Report
This is an attractive, detailed study focused on developing a novel method for design of trabecular-like porous structure (TLPS), based on the study of mechanical and permeability properties of natural bone. Finally, the created TLPS showed even better mechanical and permeability properties of natural bone.
My objections are more technical in nature:
1. Many parts in the "Results and Discussion" section, which should be concise and clear, are unnecessarily long, and might be difficult to understand, for example- lines 425-428, instead of: When C is 0.4-0.5, the maximum value is 580Mpa, and when C is 0.4-0.9, the minimum value is 300Mpa, and the compressive strength decreases by 34.9%, you can write: When C is 0.4-0.5 the compressive strength reaches the maximum value of 580Mpa, while with C value of 0.4-0.9 he compressive strength decreases by 34.9%. There is no need to repeat every value that already can be read from the figure/graph.
Please make such corrections in the whole Results and Discussion section.
2. Please add a legend to the Table 1, explaining the C, T, F, N, R
3. In lines 230, and 232 you wrote LTPS instead TLPS
4. In the line 50 you use the abbreviation SLM for the first time in the text, and in the line 70 you gave a full term, please put the full term in line 50
5. Lines 77-80- unclear sentence, please rephrase
6. In the conclusion section line 660- you mention maxillofacial "Surgery", make it more clear- i.e. complex models used in maxillofacial surgery
7. There are a lot of typos, i.e. in the Figure 2 caption TLPS "proe" size, please mace corrections through the whole text
8. English must be better, an English native speaker should revise the text
